

# Renormalization group consistency and low-energy effective theories

**Jens Braun[1,2], Marc Leonhardt[1] and Jan M. Pawlowski[2,3]**

**1** Institut für Kernphysik (Theoriezentrum),
Technische Universität Darmstadt, D-64289 Darmstadt, Germany
**2** ExtreMe Matter Institute EMMI, GSI, Planckstraße 1, D-64291 Darmstadt, Germany
**3** Institut für Theoretische Physik, Universität Heidelberg,
Philosophenweg 16, D-69120 Heidelberg, Germany

## Abstract

Low-energy effective theories have been used very successfully to study the low-energy limit of QCD, providing us with results for a plethora of phenomena, ranging from bound-state formation to phase transitions in QCD. These theories are consistent quantum field theories by themselves and can be embedded in QCD, but typically have a physical ultraviolet cutoff that restricts their range of validity. Here, we provide a discussion of the concept of renormalization group consistency, aiming at an analysis of cutoff effects and regularization-scheme dependences in general studies of low-energy effective theories. For illustration, our findings are applied to low-energy effective models of QCD in different approximations including the mean-field approximation. More specifically, we consider hot and dense as well as finite systems and demonstrate that violations of renormalization group consistency affect significantly the predictive power of the corresponding model calculations.

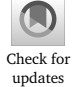

# 1 Introduction

The computation of quantum corrections in field theories in general requires a regularization and renormalization procedure. In perturbation theory, the regularization procedure allows us to compute the perturbative loop diagrams in a well-defined fashion, e.g., by introducing a momentum cutoff $\Lambda$ for the momentum integrals. This cutoff dependence can be absorbed in counter terms in the underlying bare action, here called $\Gamma_\Lambda$. The latter then consists of all ultraviolet (UV) relevant terms allowed by the symmetry of the classical theory or classical action $S$. Potentially, additional terms have to be introduced in $\Gamma_\Lambda$, if the momentum cutoff breaks symmetries present in the classical action. Here, a prominent example is provided by gauge symmetries that are explicitly broken by a momentum cutoff. Then, symmetry-breaking counter terms such as a mass term for the gauge field in $\Gamma_\Lambda$ indeed restores the gauge symmetry for the full quantum effective action.

This perturbative reasoning extends to the general non-perturbative case. In the past decade non-perturbative functional methods, such as the functional renormalization group (FRG), Dyson-Schwinger equations (DSE) and $n$PI methods, have made rapid progress and improved our understanding of strongly-correlated systems, ranging from condensed matter over heavy-ion physics and high energy-physics to quantum gravity. Inherent to all these functional approaches is their formulation in terms of non-perturbative loop equations being structurally very similar to the perturbative setting briefly discussed above. In most cases, and in particular in numerical applications, these approaches feature momentum cutoffs for the non-perturbative loops involved as well as respective counter terms in the bare action $\Gamma_\Lambda$. The explicit cutoff dependence of $\Gamma_\Lambda$ ensures the cutoff independence of the full quantum effective action $\Gamma$,

$$\Lambda\frac{d\Gamma}{d\Lambda} = 0\,. \tag{1}$$

This is the requirement of a consistent regularization and renormalization of a given theory, and is called *RG consistency*. As a central ingredient in a non-perturbative functional setup, RG consistency will be discussed in detail in Sec. 2. Evidently, these considerations are very general and are not bound to perturbatively renormalizable theories or to a specific class of field theories.

In this work, we discuss how cutoff artefacts can be removed consistently within a given low-energy effective theory in order to ensure the important property of RG consistency (1). Irrespective of possible fundamental UV completions, the discussion of cutoff artefacts is required for a meaningful application of a given model and a test of its range of applicability in terms of external parameters. In Sec. 2, we therefore discuss this issue on very general grounds. In Sec. 3, we then demonstrate the application of these general considerations to specific model calculations. This includes a quark-meson model in the vacuum limit, a diquark model at finite density, a quark-meson-diquark model at finite temperature and density, and the quark-meson model confined in a finite box. Our conclusions can be found in Sec. 4.

# 2 RG consistency

In this section we focus on general aspects of RG consistency which includes both discussions of formal as well as phenomenological aspects. The reader may skip this section in a first reading and readily start with Sec. 3 where we exemplify the meaning of RG consistency in the context of specific QCD low-energy effective theories.

## 2.1 RG consistency and low-energy phenomenology

The computation of quantum corrections in field theories in general requires the introduction of a UV cutoff $\Lambda$. This scale is said to be asymptotically large when

$$\frac{s_i}{\Lambda} \ll 1 \quad \text{with} \quad s = \{m_{\text{phys}}, m_{\text{ext}}\}, \tag{2}$$

where the set $s$ stands for all mass scales in the theory, including dimensionful couplings. In particular, this set consists of the intrinsic fundamental parameters of the theory, $m_{\text{phys}}$ (e.g. masses of particles), as well as the external scales $m_{\text{ext}}$. For example, for the low-energy effective theories (LEFT) of QCD discussed below, we have $m_{\text{ext}} = \{T, V^{-1/3}, \mu\}$, where $T$ is the temperature, $V$ is the volume of the system, and $\mu$ is the quark chemical potential.

When Eq. (2) is ensured, the bare action $\Gamma_\Lambda$ only encodes the microphysics of the situation at hand, and changes of the intrinsic parameters are simply triggered by changing the respective bare parameters in the action. In particular, Eq. (2) entails that a change of the external parameters of the theory does not change the regularization and renormalization of the theory encoded in the $\Lambda$-dependence of $\Gamma_\Lambda$. For $m_{\text{ext},i}/\Lambda \to 0$, the corresponding property then reads

$$\frac{d}{dm_{\text{ext},i}} \left[ \Lambda \frac{d\Gamma_\Lambda}{d\Lambda} \right] = 0, \tag{3}$$

which highlights the similarity of this condition to the RG-consistency condition given in Eq. (1). In turn, if Eq. (2) does not hold, $\Gamma_\Lambda$ has to vary with a change of $m_{\text{ext}}$ to ensure that the RG-consistency condition (1) holds. However, a dependence of $\Gamma_\Lambda$ on the external parameter then implies

$$\frac{d}{dm_{\text{ext},i}} \left[ \Lambda \frac{d\Gamma_\Lambda}{d\Lambda} \right] \neq 0. \tag{4}$$

This is elaborated below. Note that, if Eq. (3) is violated, (part of) the physics related to the fluctuation physics of the respective external parameters is already carried by the bare action $\Gamma_\Lambda$. It has to be computed separately, which necessitates an explicit expression for the right-hand side of Eq. (4). This computation is of eminent importance. As we shall exemplify in this work, violations of RG consistency may indeed significantly spoil predictions for physical observables.

For a plethora of physically interesting theories, the cutoff $\Lambda$ may be limited by a validity bound. A strict bound is present, if the effective theory at hand cannot be extended beyond a certain UV scale. For example, a Landau pole at the scale $\Lambda_{\text{UV}}$ is such a strict bound. Then, we have to choose $\Lambda \leq \Lambda_{\text{UV}}$. This situation applies to most effective theories for the low-energy regime of QCD, such as Nambu–Jona-Lasinio-type models (NJL) and quark-meson-type models (QM) with or without Polyakov-loop extensions [1–49], and it also applies to quantum electrodynamics and a variety of condensed-matter models.

A further, qualitatively different, validity bound of LEFTs is related to the fact, that they typically lack some of the microscopic degrees of freedom that are relevant at momentum scales $\Lambda > \Lambda_{\text{phys}}$. Then, Eq. (2) may hold for a given LEFT but, beyond the scale $\Lambda_{\text{phys}}$, the LEFT lacks the dynamics associated with the fundamental microscopic degrees of freedom. Consequently, such a LEFT cannot describe the physics at hand beyond $\Lambda_{\text{phys}}$. For example, in conventional QCD low-energy effective theories, the gluon dynamics is missing. These LEFTs describe QCD solely in terms of hadronic degrees of freedom which can only hold true for low momentum scales.

Of course, by definition, a determination of the scale $\Lambda_{\text{phys}}$ is involved as it requires an actual study of the fundamental dynamics at all momentum scales. Within the FRG approach to fundamental QCD [50–56], however, it has been shown in various studies that

the gluonic sector of QCD at low baryon-density decouples from the matter sector at scales $\Lambda_{\mathrm{phys}} \sim 0.4 \ldots 1\,\mathrm{GeV}$, see e.g. [36, 38, 57–60]. In this context, it should be noted that the scales $\Lambda$, $\Lambda_{\mathrm{UV}}$, and $\Lambda_{\mathrm{phys}}$ depend on the chosen regularization scheme, and are only related to physical momentum scales by the renormalization procedure.

In our discussion of specific models in Sec. 3, we do not aim at a determination of their values of $\Lambda_{\mathrm{phys}}$ but rather aim at a discussion of how cutoff artefacts can be removed consistently within a given model study. Irrespective of possible fundamental UV completions, such a discussion of cutoff artefacts is required for a meaningful application of a specific model within given ranges for the external parameters, in particular in the absence of an accurate knowledge of the scale $\Lambda_{\mathrm{phys}}$.

## 2.2 Quantum effective action and regularization

The central object of our general discussion is the quantum effective action $\Gamma[\Phi]$ of a given theory with field content $\Phi = (\Phi_1, \Phi_2, \ldots)^T$ in $d$ Euclidean space-time dimensions. It is the quantum analogue of the classical action and its saddle points are solutions of the quantum equations of motion (EoM). Its $n$th field derivatives, evaluated at the minimal quantum EoM, are the one-particle-irreducible (1PI) parts of the $n$-point correlation functions of the theory, $\Gamma^{(n)}[\Phi] = \langle \Phi_{i_1} \cdots \Phi_{i_n} \rangle_{\mathrm{1PI}}$ for $n > 2$. For the two-point function we have $\Gamma^{(2)}[\Phi] \cdot \langle \Phi_{i_1} \Phi_{i_2} \rangle_{\mathrm{1PI}} = 1$. In a functional approach, the effective action has the generic representation

$$\Gamma[\Phi] = \mathcal{D}_\Lambda[\Phi] + \Gamma_\Lambda[\Phi], \tag{5}$$

where $\mathcal{D}_\Lambda$ stands for all momentum-loop diagrams evaluated in the presence of the momentum cutoff $\Lambda$. This cutoff leads to finite diagrams, as momentum fluctuations with $p^2 \gtrsim \Lambda^2$ are suppressed in $\mathcal{D}_\Lambda$. Hence, these fluctuations must reside in $\Gamma_\Lambda$.

The relation (5) together with the RG-consistency condition (1) is simply the requirement that the $\Lambda$-dependence of the loops is cancelled by that in the bare action $\Gamma_\Lambda$. Moreover, we can shift the fluctuation information contained in $\mathcal{D}_\Lambda[\Phi]$ to $\Gamma_\Lambda$ by lowering the scale $\Lambda$. Indeed, we have $\lim_{\Lambda \to 0} \mathcal{D}_\Lambda = 0$ and $\lim_{\Lambda \to 0} \Gamma_\Lambda = \Gamma$.

For this interpretation, $\Gamma_\Lambda$ has to be seen as an effective action that misses the infrared dynamics of the theory carried by the diagrams $\mathcal{D}_\Lambda[\Phi]$. Hence, the UV-cutoff $\Lambda$ in the diagrams serves as an infrared (IR) cutoff $k = \Lambda$ for the scale-dependent effective action $\Gamma_k$:

$$k \partial_k \Gamma_k[\Phi] = \mathcal{F}_k[\Phi], \tag{6}$$

where $\mathcal{F}_k = -k \partial_k \mathcal{D}_k$. This fruitful block-spinning perspective is taken for FRG approaches. In its modern form, Eq. (6) is a simple one-loop equation, the Wetterich equation [61] with

$$\mathcal{F}_k[\Phi] = \frac{1}{2} \mathrm{Tr} \frac{1}{\Gamma_k^{(2)}[\Phi] + R_k} k \partial_k R_k. \tag{7}$$

The trace in (7) sums over momentum, space-time and internal indices as well as species of fields. The latter includes a minus sign for fermionic degrees of freedom as known from perturbation theory. The regulator function $R_k$ depends on the IR cutoff scale $k$ and defines the regularization scheme. It adds to the full two-point function of the regularized theory and changes the dispersion. In the IR limit, it acts as a mass and thus suppresses the IR momentum fluctuations in $\Gamma_k$. For UV momenta $p^2 \gtrsim k^2$, it decays sufficiently fast in order to keep the UV physics unchanged. Let us discuss this here at the example of a scalar field. To this end, we parameterize the regulator as

$$R_k(p^2) = p^2 r(x), \tag{8}$$

with $x = p^2/k^2$ and a dimensionless shape function $r$ determining the IR and UV asymptotics. The prefactor $p^2$ carries the classical dispersion of the scalar field. More elaborated choices substitute the latter with the momentum-dependent part of the full inverse two-point function $p^2 \to \Gamma_k^{(2)}$, known as RG- or spectrally adjusted regulators [62–66].

In general, an admissible regulator $R_k$ has to obey certain conditions regarding its behavior in the low- and large-momentum limit. For example, it has to render the momentum part of the trace in (7) finite in the UV limit and, by providing a mass gap for the fields, also in the IR limit. Hence, from an RG point of view, the regulator function specifies the Wilsonian momentum-shell integration, such that the right-hand side of the differential equation (6) is dominated by fluctuations with momenta $|p| \sim k$. It should be added that fast decays of $r(x)$ improve the convergence of the approximation scheme used, for details see Refs. [66, 67]. A common approximation scheme that is also used in the present work, the derivative expansion, is based on the expansion in powers of momenta. The applicability of this scheme to any order requires shape functions that decay faster than any polynomial in $x$. In summary, exponential or even compact support regulators are best suited for common systematic approximation schemes, ranging from the derivative expansion to vertex expansions as used in QCD.

In the set of diagrams $\mathcal{D}_k[\Phi]$, the cutoff $k$ acts as a UV cutoff. UV suppression is achieved by the occurrence of

$$\frac{1}{\Gamma_{k=0}^{(2)}[\Phi]} - \frac{1}{\Gamma_k^{(2)}[\Phi] + R_k} \tag{9}$$

for internal lines. This pattern is easily seen by integrating the flow equation (6) with (7) within one-loop perturbation theory. On the right-hand side of the flow equation the bare action enters with $\Gamma_\Lambda = S$ with $S$ being the classical action. For a sharp cutoff and constant background fields, we find

$$\frac{1}{S^{(2)}} - \frac{1}{S^{(2)} + R_k} = \frac{1}{S^{(2)}} \theta(k^2 - p^2). \tag{10}$$

Additional subtractions occur in perturbation theory by iteratively generating higher loop orders by re-inserting the result on the right-hand side of the flow equation, see e.g. Refs. [66, 68, 69]. In total, the flow equation (6) with Eq. (7) leads to a generalized Bogoliubov-Parasiuk-Hepp-Zimmermann (BPHZ)-type regularization scheme: the regularization is achieved by subtraction.

## 2.3 RG consistency – formal discussion

The effective action $\Gamma$ is obtained from Eq. (5) by integrating Eq. (6) from the initial UV scale $k = \Lambda$ to $k = 0$. For finite $k$, we find

$$\Gamma_k[\Phi] = \Gamma_\Lambda[\Phi] + \int_\Lambda^k \frac{\mathrm{d}k'}{k'} \mathcal{F}_{k'}[\Phi], \tag{11a}$$

which leads to (5) for $k \to 0$. The RG consistency condition (1) follows immediately for any $k \neq \Lambda$ from Eq. (11a) by taking the $\Lambda$-derivative.[1] We have

$$\Lambda \partial_\Lambda \Gamma_k[\Phi] = \Lambda \partial_\Lambda \Gamma_\Lambda[\Phi] - \mathcal{F}_\Lambda[\Phi] = 0, \tag{11b}$$

where we have used (6) in the last step. Note that in Eqs. (11a) and (11b) the scale $\Lambda$ is not necessarily the largest scale possible in the theory at hand, i.e. $\Lambda_{\mathrm{UV}}$. It is only some scale at which we fix the couplings of the theory.

---

[1]Note that, within the standard convention of the FRG approach, the partial derivative with respect to $\Lambda$ corresponds to the total derivative in Eq. (1).

The seemingly simple relations (11a)-(11b) offer a lot of information that is in general more difficult to access in other approaches. First of all, Eqs. (11a)-(11b) entail that RG consistency and hence cutoff independence of a theory, fundamental or effective, follows trivially in the FRG approach: the effective action at the initial scale $k = \Lambda$ has to obey the flow equation, if we vary the initial scale. Moreover, its $\Lambda$-dependence is easily extracted for large initial cutoff scales $\Lambda$ with the aid of Eq. (2). In this case, $\Gamma_\Lambda$ can be expanded in powers of $\Lambda$ with

$$\Gamma_\Lambda[\Phi] = \sum_{n \leq N_{\max}} \tilde{\gamma}_n[\Phi] \Lambda^n + \tilde{\gamma}_{\log}[\Phi] \ln \frac{\Lambda}{s_0}, \tag{12a}$$

where the term with $n = 0$ carries the physics part of the initial condition at the scale $\Lambda$. Here, we have normalized the logarithmic term with some physical scale $s_0 \in s$, e.g. the physical mass gap of the theory at hand. Choosing a different reference scale shifts terms from $\tilde{\gamma}_0$ to $\tilde{\gamma}_{\log}$. Note that the $\tilde{\gamma}_n$'s can be a collection of different field-dependent terms with the same $\Lambda$-behavior. The right-hand side of the flow equation can also be expanded in powers of $\Lambda$, and the expansion coefficients only depend on the shape function $r(x)$ and $\tilde{\gamma} = \{\tilde{\gamma}_{N_{\max}}, \tilde{\gamma}_{N_{\max}-1}, ..., \tilde{\gamma}_{\log}, \tilde{\gamma}_0, \tilde{\gamma}_{-1}, \dots\}$,

$$\mathcal{F}_\Lambda[\Phi] = \sum_{n \leq N_{\max}} f_n[\Phi; \tilde{\gamma}, r] \Lambda^n. \tag{12b}$$

Inserting Eqs. (12a) and (12b) into the flow equation (6) leads to

$$\tilde{\gamma}_{n \neq 0} = \frac{1}{n} f_n[\Phi, \tilde{\gamma}, r], \qquad \tilde{\gamma}_{\log} = f_0[\Phi, \tilde{\gamma}, r], \tag{12c}$$

where we have used $\Lambda \partial_\Lambda \Lambda^n = n \Lambda^n$, $\Lambda \partial_\Lambda \ln \Lambda = 1$. Note that there is no relation for $\tilde{\gamma}_0$ as it contains the physics input. Nonetheless, $\tilde{\gamma}_0$ appears on the right-hand side of the relations for the $\tilde{\gamma}_{n \neq 0}$ and $\tilde{\gamma}_{\log}$.

The set of relations (12c) can be solved recursively and provides the intial effective action in a well-defined and practically applicable way. Note that only a finite number of terms matter due to the $\Lambda$-suppression of the rest. The relations (12a)-(12c) also make apparent that, for asymptotically large values of $\Lambda$, the initial effective action is nothing but the bare action for the given FRG scheme. As such, it depends explicitly on the cutoff $\Lambda$, see e.g. [66, 70].

The setting above is the standard one for perturbatively renormalizable theories in the absence of Landau poles. Strictly speaking, the formulation above only applies to asymptotically free theories. In QCD, for example, we are in the fortunate situation that this simple setting applies. In general LEFTs, we typically have to deal with the existence of an actual finite UV extent given in form of a maximal UV cutoff scale $\Lambda_{\mathrm{UV}}$ due to an instability of the theory, or a phenomenologically existing UV extent $\Lambda_{\mathrm{phys}}$ above which a given LEFT does no longer provide a valid description of a more fundamental theory. A priori, a safe choice is then

$$\Lambda \leq \Lambda_{\max}, \tag{13}$$

where $\Lambda_{\max} \in \{\Lambda_{\mathrm{phys}}, \Lambda_{\mathrm{UV}}\}$. For such a choice, $\Lambda$ may not be sufficiently large compared to the external parameters $m_{\mathrm{ext}}$ of interest and we are left with the situation as described by Eq. (4). Moreover, the intrinsic scales may not even be small compared to $\Lambda$.[2] Then, the determination of the initial effective action $\Gamma_\Lambda$ with Eqs. (12a)-(12c) is no longer possible: for low initial scales $\Lambda$, the initial effective action is a complicated object itself.

In LEFTs of QCD, for example, this issue may potentially be surmounted by computing $\Gamma_\Lambda$ with the aid of RG studies of the fundamental theory, see e.g. Refs. [38, 59, 60, 71]. However,

---

[2]In the following we focus on the external parameters for clarity. However, the discussion can be straightforwardly generalized to the case of intrinsic scales.

if a sufficiently accurate determination of $\Gamma_\Lambda$ from a more fundamental theory is not available, we still have to ensure that cutoff artefacts associated with a specific choice for the scale $\Lambda$ are suppressed or even removed in our model study. Otherwise, over a wide range of the external parameters, such a model study may only resolve peculiarities of the underlying regularization schmeme. In this case, we have to make use of "pre-initial" flows that provide a systematic determination of the effects of the violations described by Eq. (4) by an RG-consistent UV completion of the LEFT at hand.

To illustrate this, let us assume that we know the effective action at some scale. In case of QCD models, for example, the effective action is often chosen to assume a simple quadratic form at some scale. In the following, this scale is denoted as $\Lambda'$, see also Sec. 3 for a discussion of specific models. The UV completion $\Gamma_\Lambda$ of the LEFT is then obtained by following the RG flow from $k = \Lambda' < \Lambda$ to $k = \Lambda \leq \Lambda_{\text{UV}}$, such that we have $\Lambda \partial_\Lambda \Gamma_\Lambda \to 0$ for $m_{\text{ext},i}/\Lambda \ll 1$, i.e. RG consistency is ensured in the presence of finite external parameters. This is demonstrated in Sec. 3 for specific models and exploits the fact that the effective action $\Gamma_\Lambda$ can be determined from Eq. (11) as

$$\Gamma_\Lambda[\Phi; m_{\text{ext}}^{(0)}] = \Gamma_{\Lambda'}[\Phi; m_{\text{ext}}^{(0)}] - \int_\Lambda^{\Lambda'} \frac{\mathrm{d}k'}{k'} \mathcal{F}_{k'}[\Phi; m_{\text{ext}}^{(0)}], \tag{14}$$

with $\Lambda$ chosen such that $m_{\text{ext},i}/\Lambda \ll 1$ for all parameters of interest. Here, $\mathcal{F}_{k'}$ depends on $\Phi$ and $m_{\text{ext}}^{(0)}$, the latter denoting a given set of "benchmark values" for the external parameters at which $\Gamma_\Lambda$ has been fixed with the aid of some set of physical low-energy observables. Typical benchmark values are the vacuum values of the external parameters. For QCD, this is vanishing temperature, infinite volume, and vanishing quark chemical potential, see also our examples in Sec. 3.

If not indicated otherwise, we shall assume from now on that $\Gamma_\Lambda$ has been fixed in the limit of vanishing external parameters. From our choice (14), we then deduce that the effective action $\Gamma_k$ remains unchanged in this limit:

$$\Gamma_k[\Phi; m_{\text{ext}}^{(0)}] = \Gamma_{\Lambda'}[\Phi; m_{\text{ext}}^{(0)}] + \int_{\Lambda'}^k \frac{\mathrm{d}k'}{k'} \mathcal{F}_{k'}[\Phi; m_{\text{ext}}^{(0)}] = \Gamma_\Lambda[\Phi; m_{\text{ext}}^{(0)}] + \int_\Lambda^k \frac{\mathrm{d}k'}{k'} \mathcal{F}_{k'}[\Phi; m_{\text{ext}}^{(0)}], \tag{15}$$

where $k < \Lambda'$. However, note that the $\Phi$-dependence of $\Gamma_\Lambda$ and $\Gamma_{\Lambda'}$ is in general different. At the same time, the choice (14) allows us to ensure $\Lambda \partial_\Lambda \Gamma_\Lambda \to 0$ for $m_{\text{ext},i}/\Lambda \to 0$, see also below. Indeed, the condition $m_{\text{ext},i}/\Lambda \to 0$ ensures that Eq. (3) is fulfilled for $\Gamma_\Lambda$. Eq. (14) also offers a practical way to compute the dependence of $\Gamma_{\Lambda'}$ on the external parameters. In other words, the chosen UV completion in form of $\mathcal{F}_{k>\Lambda'}$ has to ensure the overall consistency of the LEFT, and in particular the thermodynamical consistency. Of course, this procedure is very general and also applies to the case of asymptotically free theories as well as to asymptotically safe theories where $\Lambda_{\text{UV}}$ is infinite.

For the construction of $\Gamma_\Lambda$ in case of LEFTs with $\Lambda_{\text{phys}} < \Lambda_{\text{UV}}$, it may even be required to choose $\Lambda > \Lambda_{\text{phys}}$. At first glance, this appears to be in contradiction to the very definition of the scale $\Lambda_{\text{phys}}$. Strictly speaking, this is correct and an extension of LEFTs beyond $\Lambda_{\text{phys}}$ does not carry the physical fluctuation dynamics of the underlying fundamental theory for scales $\Lambda > \Lambda_{\text{phys}}$. Nevertheless, we may have to choose $\Lambda > \Lambda_{\text{phys}}$ in order to suppress cutoff artefacts, i.e. the failure of Eq. (3). For the generic flow equation (6), the change of the initial condition reads

$$\Lambda \frac{\partial^2 \Gamma_\Lambda[\Phi, m_{\text{ext}}]}{\partial m_{\text{ext},i} \partial \Lambda} = \frac{\partial \mathcal{F}_\Lambda[\Phi, m_{\text{ext}}]}{\partial m_{\text{ext},i}}. \tag{16}$$

Integrating (16) from $m_{\text{ext},i}^{(0)}$ to $m_{\text{ext}}$ leads to an even more convenient form,

$$\Lambda\partial_\Lambda\Gamma_\Lambda[\Phi;m_{\text{ext}}]-\Lambda\partial_\Lambda\Gamma_\Lambda[\Phi;m_{\text{ext}}^{(0)}]=\mathcal{F}_\Lambda[\Phi;m_{\text{ext}}]-\mathcal{F}_\Lambda[\Phi;m_{\text{ext}}^{(0)}]. \tag{17}$$

If Eq. (3) holds, then the initial effective action is not changed apart from its explicit dependence on $m_{\text{ext}}$. The same holds for the flow equation itself. Accordingly, if Eqs. (16) and (17) are non-vanishing for a fixed initial effective action $\Gamma_\Lambda$, then the (pre-)initial flow – and hence the initial effective action – has to change for the RG-consistency condition (1) to hold: with the representation of $\Gamma$ as the integrated flow, see Eqs. (11) and (16), we are immediately led to the RG consistency condition (1). In turn, assuming (3), and using the representation (11) of $\Gamma$ as the integrated flow, we arrive at the important constraint

$$\Lambda\partial_\Lambda\Gamma[\Phi;m_{\text{ext}}]=-\left(\mathcal{F}_\Lambda[\Phi;m_{\text{ext}}]-\mathcal{F}_\Lambda[\Phi;m_{\text{ext}}^{(0)}]\right)\stackrel{!}{=}0. \tag{18}$$

Here, the first term on the right-hand side arises from the $\Lambda$-derivative of the integrated flow (11), whereas the second term originates from the $\Lambda$-derivative of the initial effective action that is kept at its benchmark values for the external parameters. Note that Eq. (18) has been used in Ref. [72] for defining the "thermal range" $\Lambda_T[r]\equiv\Lambda[r;T]$ being the minimal cutoff value for which Eq. (18) holds to a given accuracy. From our specific examples presented in Sec. 3, it is moreover possible to extract the "density range" $\Lambda[r;\mu]$ and the "volume range" $\Lambda[r;V]$. Of course, the actual values of these quantities depend on the regularization scheme specified by the regulator shape function $r$.

More generally speaking, the external parameters set the minimal value $\Lambda[r;m_{\text{ext}},m_{\text{ext}}^{(0)}]$ of the cutoff for which Eq. (18) holds to a given accuracy. For the standard benchmark defined by choosing the vacuum values for the external parameters, the third variable can be dropped. For a given LEFT with a maximal physical UV range $\Lambda_{\text{phys}}$, this entails that only results with $m_{\text{ext}}$ in the set $\mathcal{M}_{\text{ext}}$,

$$\mathcal{M}_{\text{ext}}(m_{\text{ext}}^{(0)})=\left\{m_{\text{ext}}\,|\,\Lambda[r;m_{\text{ext}},m_{\text{ext}}^{(0)}]\leq\Lambda_{\text{phys}}[r]\right\}, \tag{19}$$

are fully trustworthy. We emphasize that all the above cutoff scales naturally depend on the regularization scheme as defined by the choice for the regulator function $r$. In turn, the set $\mathcal{M}_{\text{ext}}$ should not depend on $r$, but may be $r$-dependent in given low-level approximations.

Provided that $\Lambda_{\text{phys}}$ is known for a LEFT at hand, the set $\mathcal{M}_{\text{ext}}$ defines the physics range of this LEFT. Interestingly, this discussion makes also clear that the physics range for the external parameters depends on the chosen benchmark value for the external parameters. Of course, the latter cannot be chosen freely, as the parameters in the initial effective action $\Gamma_\Lambda[\Phi;m_{\text{ext}}^{(0)}]$ are fixed with the aid of observables at $m_{\text{ext}}^{(0)}$. Only $m_{\text{ext}}^{(0)}$, for which these observables are known, can be used as a benchmark. Still, this suggests to use available first-principles result from lattice or functional studies at finite temperature and small chemical potential with $m_{\text{ext}}^{(0)}\neq 0$ as a benchmark instead of the vacuum values. In case of QCD, this in principle allows for more reliable LEFT computations of, e.g., finite-density effects, and is pursued within "QCD-assisted" LEFTs.

Irrespective of the knowledge of $\Lambda_{\text{phys}}$ and the corresponding physics range of the LEFT under consideration, it is still crucial to use the strategy associated with Eq. (14) to remove or at least suppress cutoff artefacts in the results for physical observables within a given LEFT study. In Sec. 3, we illustrate this strategy in detail with the aid of low-energy models of QCD.

In summary, the RG consistency condition (1) of a given theory is in general a non-trivial constraint on the initial effective action at finite external parameters if Eq. (4) applies to this theory. In the present FRG framework, this is practically accessible via Eqs. (16) and (17). Moreover, the formal discussion in the present section leaves us with a practical toolbox for amending computations of observables in the presence of finite external parameters. In any case, we note that the initial effective action is non-trivial if Eq. (2) does not hold.

# 3 RG Consistency – Examples

In this section, we apply our general discussion of RG consistency to QCD low-energy models with $N_f = 2$ quark flavors and $N_c = 3$ colors. In the past, low-energy effective theories including part of the quantum, thermal and density fluctuations have been studied to an increasing level of sophistication. A rather large, but not complete, list of LEFTs ranges from NJL- and QM-type models [1–21] over quark-meson-diquark models [22–27] to models including baryonic degrees of freedom [41–44], and all of them may even be augmented with statistical confinement in terms of a Polyakov loop background and a corresponding Polyakov loop potential [28–40]. Eventually, these different models are nothing but different representations of the low-energy sector of QCD that emerges after the dynamical decoupling of the gluonic degrees of freedom at cutoff scales $\sim 0.4 \dots 1\,\mathrm{GeV}$. For FRG investigations of this decoupling phenomenon in fundamental QCD, see Refs. [50–56]; for more detailed discussions of emergent LEFTS and further investigations in this respect, see, e.g., Refs. [36, 38, 57–60].

A detailed discussion of this interesting embedding of LEFTs in QCD goes beyond the scope of the present work. Here, we rather aim at a discussion of how cutoff artefacts can be removed consistently within a given model study. However, it is worth emphasizing that the different LEFT representations of low-energy QCD discussed below can be all mapped into each other within self-consistent and systematic expansion schemes. Accordingly, the structural results obtained below in one of these models extend straightforwardly to all representations of low-energy QCD. In turn, the impact of truncation artefacts might be limited to the specific model under investigation.

## 3.1 Quark-meson model in the vacuum limit

We start our discussion of specific examples with a variant of the QM model as a representation of low-energy QCD in the vacuum limit. On the mean-field level, its relation to NJL-type models [1–7] is most apparent. Its classical or UV action is nothing but an NJL-type model in its partially bosonized form. In its instant form (no $\sim \partial_i \phi$-terms), the classical action of the QM-model reads

$$S = \int d^4 x \left\{ \bar{q} \Big( \partial\!\!\!/ + \tfrac{1}{2} \bar{h} (\sigma + i \vec{\tau} \cdot \vec{\pi} \gamma_5) \Big) q + \tfrac{1}{2} \bar{m}^2 \phi^2 \right\}, \tag{20}$$

where $\bar{h}$ denotes the Yukawa coupling between quarks and the scalar and pseudo-scalar mesons. The $\tau_i$'s are the Pauli matrices which couple the quark spinors $q$ in flavor space. The scalar fields $\phi^T = (\sigma, \vec{\pi})$ do not carry an internal charge, e.g. color and flavor. Phenomenologically, these scalar fields mediate the interaction between the quarks and carry the quantum numbers of the $\sigma$-meson, $\sigma \sim (\bar{q}q)$, and the pions, $\vec{\pi} \sim (\bar{q}\vec{\tau}\gamma_5 q)$, respectively.

Let us now compute the effective action of our model in a one-loop approximation where we only take into account purely fermionic loops. Of course, the effective action can be obtained in various ways. We shall employ the Wetterich equation, (6) with (7), which allows for a convenient computation of the scale-dependent effective action. For simplicity, we shall drop terms of the form $\sim (\partial_\mu \varphi)^2$ in our calculation although they are generated by purely fermionic loops. Here, $\varphi$ is the so-called classical scalar field associated with the quantum field $\phi$ appearing in the action $S$. Our approximations imply that we neglect the RG running of the wavefunction renormalization of the scalar fields as well as the running of the Yukawa coupling, i.e. we keep $\bar{h}_k$ constant, $\bar{h}_k \equiv \bar{h}$. By expanding the scalar fields $\varphi$ about a homogeneous background $\bar{\varphi}$, we then arrive at the following result for the RG-scale dependent effective action:

$$\frac{1}{V_4} \Gamma_k[\bar{\varphi}] = \frac{1}{V_4} \Gamma_{\Lambda'}[\bar{\varphi}] - 8 N_c L_k(\Lambda', \tfrac{1}{4}\bar{h}^2 \bar{\varphi}^2), \tag{21}$$

where $V_4$ is the four-dimensional volume of Euclidean spacetime, the auxiliary function $L_k$ parametrizes the loop integral (see below), and $\Lambda'$ denotes the scale at which we assume a simple form of the effective action $\Gamma_{\Lambda'}[\bar\varphi]$. Note that we do not indicate the dependence of $\Gamma_k$ on the classical quark fields corresponding to the quantum fields $q$ in the action $S$ here and in the following as they are set to zero.

The initial condition employed to solve the differential equation (6) is given by $\Gamma_{\Lambda'}[\bar\varphi]$. For example, as often done in conventional NJL/QM-type model studies, we choose

$$\frac{1}{V_4}\Gamma_{\Lambda'}[\bar\varphi] = \frac{1}{2}m_{\Lambda'}^2\bar\varphi^2 \,. \tag{22}$$

In this case, the parameters $\bar h$ and $\bar m_{\Lambda'}^2$ are then fixed such that the experimental/physical values of a given set of low-energy observables are recovered in the long-range limit from the effective action $\Gamma_{k\to 0}[\bar\varphi]$, e.g. the constituent quark mass $m_q = \frac{1}{2}\bar h|\bar\varphi_0|$ and the pion decay constant $f_\pi = |\bar\varphi_0|$. In principle, the three parameters $\bar h$, $\bar m_{\Lambda'}^2$, and $\Lambda'$ can be used to fix the constituent quark mass, the pion decay constant, and the mass of the $\sigma$-meson. In our numerical studies below, we only use $\bar m_{\Lambda'}^2$ and $\bar h$ to fix the constituent quark mass and the pion decay constant. However, our line of arguments with respect to the RG consistency criterion can also be applied to the former case. The appearance of three parameters is related to the fact that the Yukawa coupling is marginally relevant with its RG flow being governed only by a Gaußian fixed point [73, 74].

It is worth mentioning that it is not only conventional to parametrize the effective action as a quadratic form as given in Eq. (22) at some scale $\Lambda' \sim 0.4\dots 1\,\text{GeV}$. It rather mimics the form of the mesonic part of the effective action in QCD in this energy regime. Indeed, it has been found in FRG studies of fundamental QCD that mesonic self-interactions of higher orders are suppressed [50–56].

The auxiliary function $L_k$ parametrizing the loop integral in (21) is defined as

$$L_k(\Lambda, \chi) \;=\; \frac{1}{2}\int \frac{\mathrm{d}^4 p}{(2\pi)^4}\Big\{\ln(p^2(1+r_\psi)^2 + \chi)\Big|_k - \ln(p^2(1+r_\psi)^2 + \chi)\Big|_\Lambda\Big\}, \tag{23}$$

where $p^2 = p_0^2 + \cdots + p_3^2$. For $k \to 0$, Eq. (21) then corresponds to the standard mean-field result for the effective action for a general (mass-like) regularization scheme as specified by the regulator shape function $r_\psi$.

The shape function $r_\psi$ is implicitly defined via the definition of the regulator function $R_k \equiv R_k(p)$ appearing in Eq. (7). In order to preserve chiral symmetry, we choose the following general form for this function [8–10]:

$$R_k(p) = -\not{p}\, r_\psi\big(\tfrac{p^2}{k^2}\big) \,. \tag{24}$$

As also mentioned in Sec. 2, the shape function $r_\psi$ is to a large extent at our disposal [61]. For example, using the Litim or flat regulator [75–77] for an evaluation of the function $L_k$, we find

$$L_k(\Lambda, \chi) \;=\; \frac{1}{2}\int \frac{\mathrm{d}^4 p}{(2\pi)^4}\Big\{\ln(k^2 + \chi)\theta(k^2 - p^2)$$

$$+ \ln(p^2 + \chi)\theta(\Lambda^2 - p^2)\theta(p^2 - k^2)$$

$$- \ln(\Lambda^2 + \chi)\theta(\Lambda^2 - p^2)\Big\} \,. \tag{25}$$

In the long-range limit ($k \to 0$), this expression simplifies considerably (see e.g. [73]),

$$L_0(\Lambda, \chi) = \frac{1}{2}\int \frac{\mathrm{d}^4 p}{(2\pi)^4}\theta(\Lambda^2 - p^2)\big\{\ln(p^2 + \chi) - \ln(\Lambda^2 + \chi)\big\} \,. \tag{26}$$

For comparison, we also give the result for $L_k$ as obtained from a sharp regulator function which is often used in mean-field studies,

$$L_k(\Lambda, \chi) = \frac{1}{2} \int \frac{\mathrm{d}^4 p}{(2\pi)^4} \theta(\Lambda^2 - p^2) \theta(p^2 - k^2) \ln(p^2 + \chi). \tag{27}$$

As expected, this regulator function cuts off small as well as large momenta sharply. For $k \to 0$, we then have

$$L_0(\Lambda, \chi) = \frac{1}{2} \int \frac{\mathrm{d}^4 p}{(2\pi)^4} \theta(\Lambda^2 - p^2) \ln(p^2 + \chi), \tag{28}$$

which, together with Eq. (21), yields indeed the standard result for the effective action $\Gamma[\bar{\varphi}] \equiv \Gamma_{k \to 0}[\bar{\varphi}]$ in the mean-field approximation. Note also the difference in the expressions (25) and (27) for $L_k$ which can be traced back to the difference in the underlying regularization schemes. We add that the momentum integrations in Eqs. (25)-(28) can be performed analytically, if needed, see, e.g., Refs. [73, 78].

Although our ansatz for the effective action at the scale $\Lambda'$ mimics the situation in QCD, the effective action $\Gamma_{\Lambda'}$ at the scale $\Lambda'$ does not yet obey the RG-consistency condition (1). Therefore, we now apply our general line of arguments detailed in Sec. 2 to obtain an RG-consistent result for the effective action of our present model in the mean-field approximation. From our general discussion, we immediately conclude that the effective action $\Gamma \equiv \Gamma_{k \to 0}$ does not depend on the actual scale $\Lambda$ at which we fix $\Gamma_\Lambda$, provided that we adapt $\Gamma_\Lambda$ accordingly, see Eq. (15). Indeed, assuming $\Lambda > \Lambda'$ and using Eqs. (14) and (15), we obtain

$$\frac{1}{V_4} \Gamma[\bar{\varphi}] = \frac{1}{V_4} \Gamma_\Lambda[\bar{\varphi}] - 8 N_c L_0(\Lambda, \tfrac{1}{4} \bar{h}^2 \bar{\varphi}^2), \tag{29}$$

where

$$\frac{1}{V_4} \Gamma_\Lambda[\bar{\varphi}] = \frac{1}{2} \bar{m}_{\Lambda'}^2 \bar{\varphi}^2 + 8 N_c L_{\Lambda'}(\Lambda, \tfrac{1}{4} \bar{h}^2 \bar{\varphi}^2). \tag{30}$$

Note that $\Gamma_{\Lambda'}[\bar{\varphi}]$ and $\Gamma_\Lambda[\bar{\varphi}]$ obey a different dependence on the field $\bar{\varphi}$. This can be readily demonstrated for asymptotically large scales $\Lambda$. In this case, the initial effective action $\Gamma_\Lambda$ receives $\Lambda$-dependent corrections only from terms up to fourth order in the field $\bar{\varphi}$ as higher orders are suppressed by powers of $\Lambda$:

$$\frac{1}{V_4} \Gamma_\Lambda[\bar{\varphi}] = \frac{3 \bar{h}^2 \Lambda^2}{(4\pi)^2} \bar{\varphi}^2 - \frac{3 \bar{h}^4 \ln \Lambda}{(8\pi)^2} \bar{\varphi}^4 - \frac{\bar{h}^6}{(16\pi)^2 \Lambda^2} \bar{\varphi}^6 + \mathcal{O}\left(\frac{\bar{\varphi}^8}{\Lambda^4}\right). \tag{31}$$

In this expansion of Eq. (30) about $\Lambda \to \infty$, we have only kept terms depending explicitly on $\Lambda$ *and* $\bar{\varphi}$. In any case, in the long-range limit ($k \to 0$), the effective action (29) agrees identically with the one given in (21), as it should be. In particular, we find that the effective action $\Gamma$ obeys the RG consistency condition (1), that is $\Lambda \partial_\Lambda \Gamma[\bar{\varphi}] = 0$.

For a study with finite external parameters, we can now adjust $\Lambda$ such that cutoff artefacts are removed. The latter may appear if $\Lambda > \Lambda'$ has been chosen too small initially for a specific range of the considered parameter set. A priori, it may indeed be difficult to choose a suitable value for $\Lambda$. However, our line of arguments given in Sec. 2 shows how this issue can be resolved. Even more, it allows us to investigate systematically cutoff effects in the presence of external parameters since the vacuum physics is left unchanged.[3]

---

[3]Of course, it is mandatory that the vacuum contributions to the effective action as well as those arising in the presence of finite external parameters are regularized consistently, i.e. in exactly the same way, as worked out in detail in Sec. 2, see also Ref. [79] for a discussion of this issue in terms of a Polyakov-loop extended NJL model.

Before we shall demonstrate this explicitly, we stress that our line of arguments, which eventually led us to the RG consistency criterion in Sec. 2, goes qualitatively beyond what is sometimes called extended mean-field theory in the literature. In fact, the vacuum fermion loop associated with extended mean-field calculations is naturally included in an RG treatment and should anyway not be discarded in any other approach, see, e.g., Refs. [73, 78, 80] for detailed discussions of mean-field theory in the RG context and Refs. [71, 81] for approximative treatments of RG consistency in low-energy models of QCD. Moreover, it is clear from our line of arguments that manifestation of RG consistency in general requires to include the fully field-dependent fermion loop and, beyond the mean-field approximation, it even requires to include the fully field-dependent contributions from *all* loop diagrams considered in a specific calculation of the quantum effective action. This also becomes apparent from the right-hand side of the differential equation (6) which includes contributions from all fields of a given theory, e.g. by means of the Wetterich equation.

### 3.2 Diquarks – equation of state

As a second example, we consider the computation of the equation of state of a simple quark-diquark model as a function of the quark chemical potential at vanishing temperature. The model is defined by the following classical action (for reviews see, e.g., [22–25]):

$$S = \int d^4x \left\{ \bar{q}\left(\slashed{\partial} - \mu\gamma_0\right)q + \bar{\nu}^2 \Delta_A^* \Delta_A + \bar{q}\gamma_5\tau_2\Delta_A^* T^A \mathcal{C}\bar{q}^T - q^T \mathcal{C}\gamma_5\tau_2\Delta_A T^A q \right\}, \quad (32)$$

where $\mathcal{C}$ is the charge conjugation operator and the sum over the color index $A$ runs only over antisymmetric color generators $T^A$ in the fundamental representation. The complex-valued scalar fields $\Delta_A$ carry the quantum numbers of diquark states, $\Delta_A \sim (\bar{q}\gamma_5\tau_2 T^A \mathcal{C}\bar{q}^T)$, with $J^P = 0^+$. The parameter $\bar{\nu}$ is at our disposal and can be used to determine the ground-state properties of the vacuum in this model. From a general fixed-point analysis, see [82, 83] and e.g. [84] for a mean-field analysis, it follows immediately that two qualitatively distinct scenarios are possible. To be specific, we may choose $\bar{\nu}^2$ to be positive but small such that already the ground state in the vacuum limit is governed by the formation of a diquark condensate breaking the $U_V(1)$ symmetry of our model. Alternatively, we may choose a sufficiently large value of $\bar{\nu}^2$ such that the $U_V(1)$ symmetry is only broken at finite $\mu$ due to the existence of a Cooper instability in the system but remains intact in the vacuum limit. In the latter, we therefore conclude that a critical value $\bar{\nu}_*$ (associated with a non-Gaußian fixed point) exists which separates these two distinct scenarios from each other.

Let us now compute the effective action of this model in a one-loop approximation where we only take into account the purely fermionic loop again. Moreover, we set the wavefunction renormalizations associated with the diquark fields to zero. In other words, we shall drop terms of the following form in our computation of the effective action:

$$\int d^4x \left\{ \frac{1}{2} Z_\perp(|\Delta|^2)|\vec{\nabla}\Delta|^2 + \frac{1}{2} Z_\parallel(|\Delta|^2)|\partial_\tau\Delta|^2 + \mu Z_\mu(|\Delta|^2)(\Delta\partial_\tau\Delta^* - \Delta^*\partial_\tau\Delta) \right\}, \quad (33)$$

where $\Delta^*\mathcal{O}\Delta \equiv \sum_A \Delta_A^* \mathcal{O}\Delta_A$ and $|\mathcal{O}\Delta|^2 \equiv \sum_A |\mathcal{O}\Delta_A|^2$ with $\mathcal{O}$ being some operator acting on the diquark fields. Note that, in general, such terms are dynamically generated due to quantum effects, even if only purely fermionic loops are taken into account. As a consequence of the listed approximations, we also do not take into account a scale dependence of the Yukawa-type quark-diquark coupling but set it to be constant. Using the Wetterich equation (7) and expanding the diquark fields about a homogeneous background $\bar{\Delta}_A$, we then obtain the

following expression for the scale-dependent effective action:

$$\frac{1}{V_4}\Gamma_k[\{\bar{\Delta}_A\}] = \frac{1}{V_4}\Gamma_{\Lambda'}[\{\bar{\Delta}_A\}] - \frac{\mu^4}{6\pi^2} - 8M_k(\Lambda',|\bar{\Delta}|^2),\tag{34}$$

where $|\bar{\Delta}|^2 = \sum_A |\bar{\Delta}_A|^2$ and $\Lambda'$ denotes again the scale at which we know the form of the effective action. The contribution $\sim \mu^4$ arises from quark degrees of freedom which do not couple to the diquark fields and therefore appear as non-interacting "spectators".

The loop integral associated with the effective action (34) is parametrized by the function $M_k$,

$$M_k(\Lambda,\chi) = \frac{1}{2}\int\frac{\mathrm{d}^3 p}{(2\pi)^3}\sum_{\sigma=\pm 1}\left(\omega^{(\sigma)}\Big|_k - \omega^{(\sigma)}\Big|_\Lambda\right),\tag{35}$$

where the auxiliary quantity $\omega^{(\sigma)}$ may be viewed as (infrared) regularized quasiparticle energy,

$$\omega^{(\sigma)} = \sqrt{(|\vec{p}\,|(1+r_\psi)+\sigma\mu)^2 + \chi}\,.\tag{36}$$

Evidently, for $\chi = 0$, we have $\omega^{(\sigma)} = |\vec{p}\,|(1+r_\psi)+\sigma\mu$. In the calculation of the loop integral (35), we have now employed the class of so-called $3d$ regulator functions which is defined as

$$R_k(p) = -\slashed{\vec{p}}\, r_\psi(\tfrac{\vec{p}^{\,2}}{k^2})\,.\tag{37}$$

This class of regularization schemes is also frequently used in QCD model studies since it allows to perform analytically the Matsubara sums in at least some of the loop diagrams. However, it should also be noted that $3d$ regularization schemes break the Poincaré symmetry explicitly. This explicit breaking is present even in the limit of vanishing temperature and chemical potential, see, e.g., Refs. [72, 80, 82, 85, 86]. The appearance of this issue can be traced back to the fact that, by construction, $3d$ regulators do not cut off the time-like momentum modes, thereby treating the time-like and spatial modes differently. We shall ignore this issue in our present study.

For convenience, we shall only consider the $3d$ sharp cutoff in our numerical studies below. The function $M_k$ is then given by

$$M_k(\Lambda,\chi) = \frac{1}{2}\int\frac{\mathrm{d}^3 p}{(2\pi)^3}\theta(\Lambda^2-\vec{p}^{\,2})\theta(\vec{p}^{\,2}-k^2)\left\{\sqrt{(|\vec{p}\,|+\mu)^2+\chi}+\sqrt{(|\vec{p}\,|-\mu)^2+\chi}\right\},\tag{38}$$

which reduces to the standard mean-field expression in the limit $k \to 0$.

Before we present our results for the equation of state of our diquark model, we would like to discuss first a subtlety in our calculation: In contrast to a possible renormalization of the quark chemical potential driven by diagrams with internal bosonic and fermionic lines, the renormalization of the chemical potential of the diquarks associated with a term $\sim \mu^2|\Delta|^2$ is already included in our present analysis. Indeed, the field-dependent renormalization factor $Y \equiv Y_{k\to 0}$ of the diquark chemical potential is given by

$$Y_k(|\bar{\Delta}|^2) = -\frac{1}{4V_4}\partial_\mu^2\Gamma_k[\{\bar{\Delta}_A\}]\Big|_{\mu=0}\,.\tag{39}$$

Using Eq. (34) for the effective action, we find $Y \sim |\Delta|^2 \ln\Lambda' + \dots$. Thus, $Y$ exhibits the same dependence on $\Lambda'$ as expected for the renormalization factors of kinetic terms, such as the ones for the diquark fields in Eq. (33). This coincidence in the $\Lambda'$-dependence of $Y$ and, e.g., $Z_\parallel$ is by no means accidental. It is rather related to a more abstract symmetry of our model which

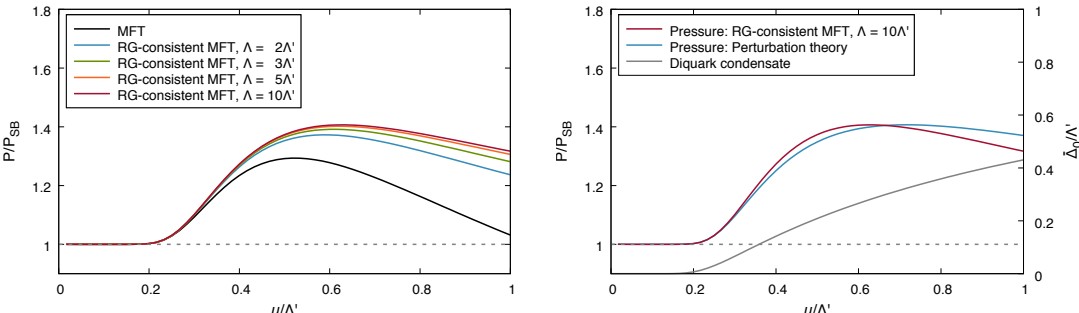

Figure 1: Left panel: Pressure $P/P_{\mathrm{SB}}$ of our diquark model as a function of the chemical potential $\mu/\Lambda'$ (with $\Lambda' = 0.6\,\mathrm{GeV}$) as obtained from conventional mean-field theory (MFT) with a UV cutoff $\Lambda = \Lambda'$ (black line) as well as from RG-consistent MFT with $\Lambda/\Lambda' = 2, 3, 5, 10$. Right panel: Pressure $P/P_{\mathrm{SB}}$ of our diquark model as a function of the chemical potential $\mu/\Lambda'$ as obtained from RG-consistent MFT with $\Lambda/\Lambda' = 10$ together with the perturbative expression for the pressure at leading order in the weak-coupling expansion, see Eq. (46). Moreover, we also show the gap $\bar{\Delta}_0/\Lambda'$ (gray line) as extracted from RG-consistent MFT with $\Lambda/\Lambda' = 10$.

is associated with the so-called Silver-Blaze property of quantum field theories [87–90]. In general, this property is linked to the fact that the free energy should not exhibit a dependence on the baryon/quark chemical potential at *zero* temperature, provided that it is smaller than some critical value. Then, the corresponding symmetry is not violated. The critical value is set by the gaps in the propagators of the fields associated with a finite baryon number. Note that the gap is not necessarily given by the physical (pole) mass. In our RG study, for example, the gap may also arise for $k > 0$ from the IR regularization of the propagator, see Ref. [89] for details.

In the presence of the symmetry associated with the Silver-Blaze property [89], a finite renormalization factor $Y$ of the diquark chemical potential implies that the renormalization factors $Z_\perp$, $Z_\parallel$, and $Z_\mu$ of the diquark fields are in principle finite as well. In mean-field calculations, these renormalization factors are usually set to zero. Therefore, the resulting effective action violates the Silver-Blaze property.[4] As already stated above, we shall not compute these renormalization factors in this work but also set them to zero. Since we shall fix the couplings/parameters of our model at a scale $k = \Lambda' > \mu$, i.e. at a point where the model is expected to respect the symmetry associated with the Silver-Blaze property, we set the initial condition for the renormalization factor $Y$ to zero as well. This ensures that this property is at least manifestly present at the scale $\Lambda'$ at which we fix the parameters of the model. To be specific, we make the following ansatz for the effective action at the scale $\Lambda'$ in the vacuum limit,

$$\frac{1}{V_4}\Gamma_{\Lambda'}[\{\bar{\Delta}_A\}] = \bar{\nu}_{\Lambda'}^2 |\bar{\Delta}|^2 \,, \tag{40}$$

where $\bar{\nu}_{\Lambda'}^2$ is at our disposal and corresponds to the parameter $\bar{\nu}^2$ in the classical action (32). However, our choice (40) for the effective action at the scale $\Lambda'$ does not imply that $Y$ remains zero at scales $k \neq \Lambda'$. Since we do not take into account the running of the renormalization factors $Z_\perp$, $Z_\parallel$, and $Z_\mu$, the symmetry associated with the Silver-Blaze property is therefore in

---

[4]Irrespective of the regularization scheme, the so-called Silver-Blaze property of the theory is already violated by the fact that the quasiparticle energies are only positive semi-definite in (standard) mean-field approximations, see Eq. (36).

general violated away from the scale $\Lambda'$. Still, the consideration of the renormalization factor $Y$ is required to ensure RG consistency within our model study, see below.

Along the lines of our discussion of the vacuum limit, we can now construct an RG-consistent effective action $\Gamma_{k\to 0}$ from (34) by adapting the effective action at the scale $\Lambda > \Lambda'$ such that the effective action at scales $k \leq \Lambda'$ remains unchanged:

$$\frac{1}{V_4}\Gamma_k[\{\bar{\Delta}_A\}] = \frac{1}{V_4}\Gamma_\Lambda[\{\bar{\Delta}_A\}] - \frac{\mu^4}{6\pi^2} - 8M_k(\Lambda, |\bar{\Delta}|^2), \tag{41}$$

where

$$\frac{1}{V_4}\Gamma_\Lambda[\{\bar{\Delta}_A\}] = \frac{1}{V_4}\Gamma_{\Lambda'}[\{\bar{\Delta}_A\}] + 8M_{\Lambda'}(\Lambda, |\bar{\Delta}|^2)\Big|_{\mu=0} + 4\mu^2\left(\partial_\mu^2 M_{\Lambda'}(\Lambda, |\bar{\Delta}|^2)\Big|_{\mu=0}\right). \tag{42}$$

Here, the last term on the right-hand side accounts for the fact that the diquark chemical potential is renormalized. Using Eq. (41), we indeed find that $\Gamma$ is RG-consistent in a strict sense in the limit $\Lambda \to \infty$ since

$$\Lambda\partial_\Lambda\Gamma[\{\bar{\Delta}_A\}] = -2V_4|\bar{\Delta}|^2\mu^2\left(\frac{\mu}{\pi\Lambda}\right)^2 + \mathcal{O}(1/\Lambda^4). \tag{43}$$

Moreover, we deduce from Eq. (41) that the renormalization of the diquark chemical potential still vanishes identically at the scale $\Lambda'$,

$$Y_{\Lambda'}(|\bar{\Delta}|^2) = 0, \tag{44}$$

as it should be. With our RG-consistent effective action (41) at hand, we now compute the equation of state of our diquark model. More specifically, we compute the pressure $P$ which is directly obtained from the effective action:

$$P = -\frac{1}{V_4}\Gamma[\{\bar{\Delta}_A\}_{gs}]\Big|_\mu + \frac{1}{V_4}\Gamma[\{\bar{\Delta}_A\}_{gs}]\Big|_{\mu=0}. \tag{45}$$

Here, the subscript 'gs' indicates that the effective action is evaluated on the $\mu$-dependent minimum (i.e. on the ground-state (gs) configuration of the fields $\{\bar{\Delta}_A\}$). Note that we have normalized the pressure with respect to the pressure in the vacuum limit. The latter is given by the second term on the right-hand side of Eq. (45).

As an explicit example, we compute the pressure of our pure diquark model for $(\bar{v}_{\Lambda'}/\bar{v}_*)^2 = 4/3$ where $\bar{v}_*^2 \approx 0.036$. Moreover, we set $\Lambda' = 0.6\,\text{GeV}$ in the following. Phenomenologically speaking, our parameter choice implies that the $U_V(1)$ symmetry is only broken at finite $\mu$ but remains intact in the vacuum limit, see our discussion above. Thus, the ground state in the vacuum limit is governed by ungapped quarks.

In the left panel of Fig. 1, we show our results for the pressure $P/P_{\text{SB}}$ of our diquark model, where $P_{\text{SB}} = \mu^4/(2\pi^2)$ denotes the Stefan-Boltzmann limit of the pressure, i.e. the pressure of a free quark gas at zero temperature. We observe that cutoff effects become continuously smaller when $\Lambda/\Lambda'$ is increased. Recall that, in our RG-consistent calculations, an increase of $\Lambda$ leaves the model in the vacuum limit unchanged. Moreover, we find that the corrections to the results from the conventional mean-field study are significant. Indeed, the pressure obtained from the conventional mean-field study underestimates the (effectively) cutoff-independent result for the pressure obtained from our RG-consistent mean-field study (with $\Lambda/\Lambda' = 10$) by about 10% at $\mu/\Lambda' = 1/2$. Thus, "cutoff contaminations" are clearly visible even at values of the chemical potential which seem to be sufficiently small compared to the originally chosen scale $\Lambda'$. At $\mu/\Lambda' = 1$, the results from the conventional mean-field study and our RG-consistent mean-field study (with $\Lambda/\Lambda' = 10$) then already deviate by about 30%. Increasing $\mu$

even further, we observe that the pressure approaches the Stefan-Boltzmann limit from above, provided $\Lambda/\Lambda'$ has been chosen sufficiently large.

From (45), we can also derive the perturbative result for the pressure. At leading order of $|\bar{\Delta}_0|^2/\mu^2$ in the weak-coupling expansion, we indeed recover the well-known result [91, 92]:

$$\frac{P}{P_{\text{SB}}} = 1 + \frac{2|\bar{\Delta}_0|^2}{\mu^2} + \dots, \tag{46}$$

where $\bar{\Delta}_0$ denotes the gap as obtained from a minimization of the effective action. In the right panel of Fig. 1, we compare this perturbative result for the pressure with the results from our RG-consistent mean-field calculation with $\Lambda/\Lambda' = 10$. Moreover, the gap as obtained from the same RG-consistent calculation is shown. Plugging this result for the gap into the perturbative expression (46) for $P/P_{\text{SB}}$, we find very good agreement with the RG-consistent results for the pressure in the regime where $|\bar{\Delta}_0|/\mu \lesssim 0.5$. For larger values of $\mu$, the results from the perturbative approximation of the pressure then exceed the results from the RG-consistent calculation. Still, the perturbative expression for the pressure appears to provide us with a reasonable estimate for the pressure over a wide range of the chemical potential, at least for our present choice for the model parameter $\bar{\nu}_{\Lambda'}$.

## 3.3 Quarks, mesons, and diquarks – phase diagram

Let us now turn to our third example, the computation of the phase diagram and the zero-temperature equation of state of a quark-meson-diquark model with two massless quark flavors and $N_c = 3$ colors. The classical action $S$ underlying our study may be viewed as a combination of the actions already discussed in Subsecs. 3.1 and 3.2 and reads

$$S = \int d^4x \left\{ \bar{q}\left(\partial\!\!\!/ - \mu\gamma_0 + \frac{1}{2}\bar{h}(\sigma + i\vec{\tau}\cdot\vec{\pi}\gamma_5)\right)q + \bar{q}\gamma_5\tau_2\Delta_A^* T^A \mathcal{C}\bar{q}^T \right.$$

$$\left. - q^T \mathcal{C}\gamma_5\tau_2\Delta_A T^A q + \frac{1}{2}\bar{m}^2\phi^2 + \bar{\nu}^2\Delta_A^*\Delta_A \right\}, \tag{47}$$

where $\bar{h}$, $\bar{m}^2$ and $\bar{\nu}^2$ are parameters at our disposal. In the following, we compute the effective action of this model in a one-loop approximation where we only take into account purely fermionic loops. Moreover, the wavefunction renormalization factors of the meson and diquark fields are set to zero again. These approximations also imply that we neglect the RG runnings of the Yukawa-type couplings of our model. As before, we moreover neglect corrections to the wavefunction renormalization factors of the quark fields (as well as to the quark chemical potential). Note that our discussion in the previous subsection regarding the fate of the Silver-Blaze property in mean-field-like calculations also holds for the present study of a quark-meson-diquark model.

Using the Wetterich equation (7) with the class of $3d$ regulator functions and expanding the meson and diquark fields about homogeneous backgrounds $\bar{\varphi}$ and $\bar{\Delta}_A$, respectively, we obtain the following result for the scale-dependent RG-consistent effective action:

$$\frac{T}{V}\Gamma_k[\bar{\varphi}, \{\bar{\Delta}_A\}] = \frac{T}{V}\Gamma_\Lambda[\bar{\varphi}, \{\bar{\Delta}_A\}] - 4L_k^{(T)}(\Lambda, \tfrac{1}{4}\bar{h}^2\bar{\varphi}^2) - 8M_k^{(T)}(\Lambda, \tfrac{1}{4}\bar{h}^2\bar{\varphi}^2, |\bar{\Delta}|^2), \tag{48}$$

where $T$ is the temperature, $V$ is the spatial volume of the system, and

$$\frac{T}{V}\Gamma_\Lambda[\bar{\varphi}, \{\bar{\Delta}_A\}] = \frac{T}{V}\Gamma_{\Lambda'}[\bar{\varphi}, \{\bar{\Delta}_A\}] + 4L_{\Lambda'}^{(T)}(\Lambda, \tfrac{1}{4}\bar{h}^2\bar{\varphi}^2)\Big|_{T=\mu=0}$$

$$+ 4\mu^2\left(\partial_\mu^2 M_{\Lambda'}^{(T)}(\Lambda, \tfrac{1}{4}\bar{h}^2\bar{\varphi}^2, |\bar{\Delta}|^2)\Big|_{T=\mu=0}\right) + 8M_{\Lambda'}^{(T)}(\Lambda, \tfrac{1}{4}\bar{h}^2\bar{\varphi}^2, |\bar{\Delta}|^2)\Big|_{T=\mu=0}. \tag{49}$$

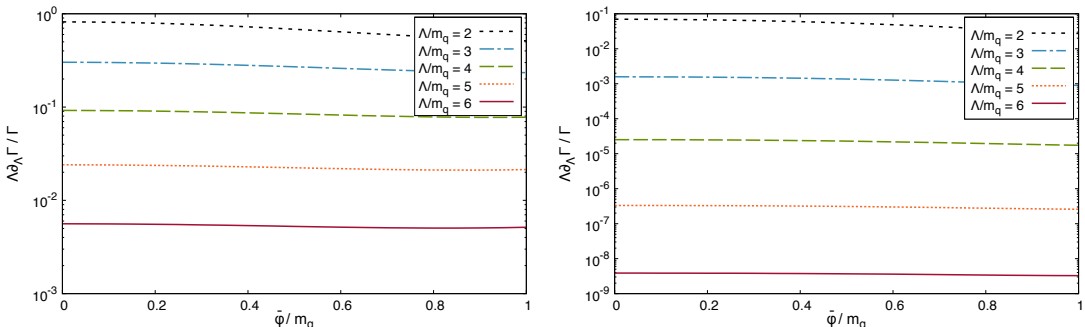

Figure 2: Change of the effective action at $\bar{\Delta} = 0$ under a variation of the UV scale $\Lambda$, i.e. $\Lambda \partial_\Lambda \Gamma$, relative to the effective action $\Gamma$ itself as a function of $\bar{\varphi}$ for $T/m_{\rm q} = 1/2$ and $\mu = 0$ (left panel) as well as for $T/m_{\rm q} = 1/5$ and $\mu/m_{\rm q} = 1$ (right panel) for various different values of $\Lambda/m_{\rm q}$, where $m_{\rm q} \approx 0.300\,{\rm GeV}$ is the vacuum quark mass.

Here, the auxiliary functions $L_k^{(T)}$ and $M_k^{(T)}$ parametrize loop integrals in the presence of a heat bath with temperature $T = 1/\beta$, see below for their definitions. The term $\sim \mu^2$ in Eq. (49) accounts for the renormalization of the chemical potential of the diquarks. As done in the previous subsections, we shall assume that the parameters of the model are fixed at the scale $k = \Lambda' < \Lambda$ by means of an ansatz for $\Gamma_{\Lambda'}$ in Eq. (49). For a study of the effect of a finite temperature and/or quark chemical potential, the scale $\Lambda$ then has to be chosen sufficiently large such that cutoff artefacts are suppressed. For $\Gamma_{\Lambda'}$, to be specific, we use the following ansatz (in the vacuum limit):

$$\lim_{T \to 0} \frac{T}{V} \Gamma_{\Lambda'}[\bar{\varphi}, \{\bar{\Delta}_A\}] = \frac{1}{2} \bar{m}_{\Lambda'}^2 \bar{\varphi}^2 + \bar{\nu}_{\Lambda'}^2 |\bar{\Delta}|^2 \,. \tag{50}$$

As often done in quark-meson-diquark model studies [22–25], we shall relate the parameters appearing in Eq. (50) via $2\bar{m}_{\Lambda'}^2/\bar{h}^2 = (3/4)\bar{\nu}_{\Lambda'}^2$ and fix $\bar{h}$, $\bar{m}_{\Lambda'}^2$ at $\Lambda'/m_{\rm q} = 2$ in the vacuum limit such that we obtain $m_{\rm q} = \frac{1}{2}\bar{h}\bar{\varphi}_0 \approx 0.300\,{\rm GeV}$ for the quark mass and $f_\pi = 2m_{\rm q}/\bar{h} \approx 0.088\,{\rm GeV}$ for the pion decay constant.

For sufficiently large values of $\Lambda$, the effective action $\Gamma_\Lambda$ receives corrections only from terms up to fourth order in the fields $\bar{\varphi}$ and $\bar{\Delta}$, respectively. Higher orders are suppressed when $\Lambda$ is increased. This resembles the situation in Subsec. 3.1. Recall that we have $\Lambda \partial_\Lambda \Gamma = 0$ by construction at $T = \mu = 0$ and, for sufficiently large values of $\Lambda$, also at $T > 0$ and/or $\mu > 0$. From Eqs. (48) and (49), however, we deduce that $\Gamma_{\Lambda'}$ depends on $T$ and $\mu$ and is no longer only quadratic in the fields for $T > 0$ and/or $\mu > 0$. This implies that RG consistency is in general violated in conventional QCD low-energy model studies with fixed $\Lambda' = \Lambda$ since these modifications of $\Gamma_{\Lambda'}$ at $T > 0$ and/or $\mu > 0$ are not taken into account. There, the quadratic form (50) is rather left unchanged for any value of the external parameters.

For convenience, we shall restrict ourselves to the $3d$ sharp regulator in our numerical calculations below. Then, the auxiliary functions parametrizing the loop integrals in Eqs. (48) and (49) read

$$L_k^{(T)}(\Lambda, \chi) = \frac{1}{2} \int \frac{{\rm d}^3 p}{(2\pi)^3} \sum_{\sigma = \pm 1} \left\{ \left( \omega_\varphi^{(\sigma)} + 2T \ln\left(1 + e^{-\beta \omega_\varphi^{(\sigma)}}\right) \right)\Big|_k \right.$$

$$\left. - \left( \omega_\varphi^{(\sigma)} + 2T \ln\left(1 + e^{-\beta \omega_\varphi^{(\sigma)}}\right) \right)\Big|_\Lambda \right\} \tag{51}$$

with

$$\omega_\varphi^{(\sigma)} = \sqrt{\vec{p}^{\,2}(1 + r_\psi)^2 + \chi} + \sigma\mu \tag{52}$$

and

$$M_k^{(T)}(\Lambda, \chi, \xi) = \frac{1}{2} \int \frac{\mathrm{d}^3 p}{(2\pi)^3} \sum_{\sigma=\pm 1} \left\{ \left( \omega_\Delta^{(\sigma)} + 2T \ln\left(1 + \mathrm{e}^{-\beta \omega_\Delta^{(\sigma)}}\right)\right) \Big|_k \right.$$
$$\left. - \left( \omega_\Delta^{(\sigma)} + 2T \ln\left(1 + \mathrm{e}^{-\beta \omega_\Delta^{(\sigma)}}\right)\right) \Big|_\Lambda \right\} \tag{53}$$

with

$$\omega_\Delta^{(\sigma)} = \sqrt{\left(\sqrt{\vec{p}^{\,2}(1 + r_\psi)^2 + \chi} + \sigma\mu\right)^2 + \xi}. \tag{54}$$

For $\xi = 0$, we use $\omega_\Delta^{(\sigma)} = \omega_\varphi^{(\sigma)}$ to preserve the Silver-Blaze property along the axis associated with $\bar{\Delta} = 0$. For the $3d$ sharp regulator function, for example, these functions are given by

$$L_k^{(T)}(\Lambda, \chi) = \frac{1}{2} \int \frac{\mathrm{d}^3 p}{(2\pi)^3} \theta(\Lambda^2 - \vec{p}^{\,2}) \theta(\vec{p}^{\,2} - k^2) \sum_{\sigma=\pm 1} \left( \omega_\varphi^{(\sigma)} + 2T \ln\left(1 + \mathrm{e}^{-\beta \omega_\varphi^{(\sigma)}}\right)\right) \Big|_{(1 + r_\psi) \to 1} \tag{55}$$

and

$$M_k^{(T)}(\Lambda, \chi, \xi) = \frac{1}{2} \int \frac{\mathrm{d}^3 p}{(2\pi)^3} \theta(\Lambda^2 - \vec{p}^{\,2}) \theta(\vec{p}^{\,2} - k^2) \sum_{\sigma=\pm 1} \left( \omega_\Delta^{(\sigma)} + 2T \ln\left(1 + \mathrm{e}^{-\beta \omega_\Delta^{(\sigma)}}\right)\right) \Big|_{(1 + r_\psi) \to 1}. \tag{56}$$

Corresponding expressions for $L_k^{(T)}$ for the $3d$ Litim regulator can be found in Ref. [73]. We note that the effective action (48) is identical to the effective action (41) for $\bar{\varphi} = 0$ in the limit $T \to 0$.

In Fig. 2, we show $(\Lambda \partial_\Lambda \Gamma)/\Gamma$, i.e. the change of the effective action under variation of the scale $\Lambda > \Lambda'$ relative to $\Gamma$ itself, at $\bar{\Delta} = 0$ as a function of $\bar{\varphi}$ for various different values of $\Lambda/m_q$. We observe that $\Gamma$ exhibits a strong dependence on our choice for $\Lambda$ in the phenomenologically most relevant regime $\bar{\varphi} \lesssim f_\pi$. In particular, this is true close to the critical temperature at $\mu = 0$, see left panel of Fig. 2, where cutoff artefacts are still clearly present in the effective action even for already seemingly large values of $\Lambda > \Lambda'$. At low temperature but large quark chemical potential $\mu \gtrsim m_q$, see right panel of Fig. 2, cutoff contaminations of the effective action are also present but appear to be less strong compared to the case with $\mu = 0$. However, this is misleading as the minimum of the effective action is pushed away from the axis with $\bar{\Delta} = 0$ in this regime. There, the dynamics is no longer governed by the pions and the $\sigma$-meson but rather by the diquark degrees of freedom. Indeed, close to the physical minimum of the effective action in this regime, cutoff effects even appear to be stronger as in the case with $\mu = 0$. This can be inferred from the phase diagram in the $(T, \mu)$ plane as well as from the pressure at zero temperature. We emphasize that the value of $\Lambda$ associated with effectively converged results depends on the temperature, the quark chemical potential, and the employed regularization scheme. Note that the effective actions associated with different values of $\Lambda > \Lambda'$ agree identically in the vacuum limit, i.e. we have $\Lambda \partial_\Lambda \Gamma = 0$ in this limit.

In Fig. 3, we present the results for the $(T, \mu)$ phase diagram of our quark-meson-diquark model. Qualitatively, the structure of the phase diagram is determined by the emergence of three different phases: a phase governed by spontaneous chiral symmetry breaking at low temperature and small quark chemical potential, a phase governed by spontaneous $U_V(1)$-symmetry breaking as associated with diquark condensation at low temperature and large chemical potential, and a symmetric high-temperature phase. Moreover, for our parameter

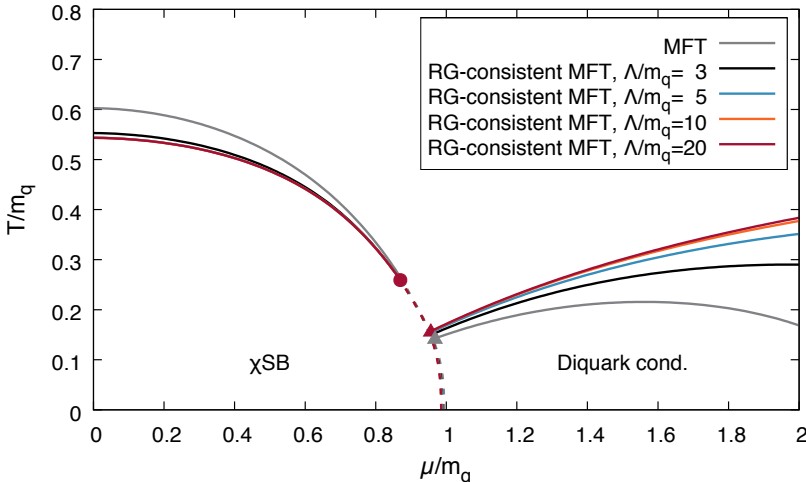

Figure 3: Phase diagram of the quark-meson-diquark model in the plane spanned by the dimensionless temperature $T/m_q$ and the dimensionless chemical potential $\mu/m_q$ for various different values of $\Lambda/m_q$ (with $m_q \approx 0.300\,\text{GeV}$). Solid lines are associated with second-order phase transitions whereas dashed lines are associated with first-order phase transitions. Note that the effective actions obtained with different values of $\Lambda$ agree identically in the vacuum limit, i.e. the RG-consistency condition (1) is strictly satisfied in this limit.

choice, we observe the existence of a critical endpoint (depicted by the dot in Fig. 3), at which the line of chiral second-order phase transitions meets a line of chiral first-order phase transitions, as well as a triple point (depicted by the triangle in Fig. 3), at which the phase governed by chiral symmetry breaking meets the diquark phase and the symmetric high-temperature phase. The general structure of the phase diagram suggests that a description of the dynamics in terms of only quarks, pions, and $\sigma$ mesons is insufficient for $T/\mu \lesssim 0.2$ and $\mu/m_q \gtrsim 1$. Below this line, diquark degrees of freedom become relevant, as well-known from previous mean-field studies [22–25]. Note that these general statements on the structure of the phase diagram are also in accordance with a recent Fierz-complete NJL model study beyond the mean-field limit [83]. Of course, in addition to the issue of an RG-consistent treatment of cutoff artefacts as discussed in our present work, artefacts from specific truncations of the effective action may become relevant in the dense and/or low-temperature regime, see, e.g., Ref. [93–95].

The general structure of the phase diagram appears to be insensitive with respect to an increase of the cutoff scale $\Lambda$, at least for the values of the model parameters used in our numerical studies. However, the positions of the two second-order phase transition lines exhibit a strong dependence on $\Lambda$, meaning that they converge only slowly when $\Lambda$ is increased, in particular at large chemical potential, see Fig. 3. To be more specific, despite the fact that we employed a 3$d$ regulator, the critical temperature at $\mu = 0$ is lowered by about 10% compared to the conventional mean-field study when we take into account cutoff corrections enforced by the RG-consistency condition (1). In the regime governed by diquark dynamics, we observe that the critical temperature is not decreased but rather increased significantly when cutoff corrections are taken into account. Compared to the conventional mean-field study (associated with $\Lambda = \Lambda'$), we indeed find a change of about 30% at $\mu/m_q \approx 4/3$ and about 100% at $\mu/m_q \approx 2$. The strength of cutoff artefacts in the high-density regime also becomes apparent in other observables, such as the pressure of the system at zero temperature as a function of the quark chemical potential, see Fig. 4. Here, we find that the pressure now exceeds the pres-

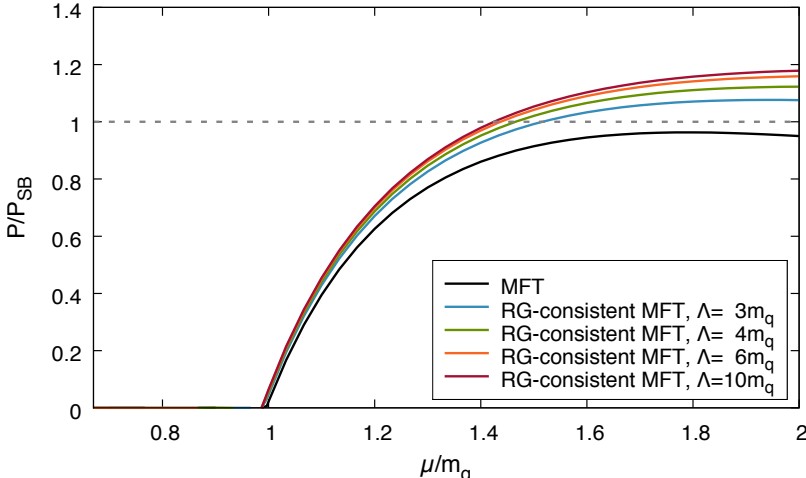

Figure 4: Pressure $P/P_{\mathrm{SB}}$ of our quark-meson-diquark model as a function of the chemical potential $\mu/m_{\mathrm{q}}$ (with $m_{\mathrm{q}} \approx 0.300\,\mathrm{GeV}$) as obtained from conventional MFT associated with $\Lambda/m_{\mathrm{q}} \equiv \Lambda'/m_{\mathrm{q}} = 2$ (black line) as well as from RG-consistent MFT with $\Lambda/m_{\mathrm{q}} = 3, 4, 6, 10$.

sure $P_{\mathrm{SB}}$ of the free quark gas once cutoff artefacts have been removed. Increasing the quark chemical potential further, we eventually observe that the pressure approaches the pressure of the free gas from above, as also observed for the pure diquark model, see Fig. 1. Clearly, it appears crucial to enforce RG consistency in the high-density regime. Note that our observations may even be very relevant from a phenomenological point of view since the associated corrections may significantly alter the equation of state of dense strong-interaction matter as relevant for astrophysical applications [96].

## 3.4 Quarks and mesons – finite-volume effects

As a fourth and final example, we demonstrate that our general line of arguments detailed in Sec. 2 can also be applied straightforwardly to studies beyond the mean-field approximation as well as to studies with an external parameter other than the temperature or the quark chemical potential. To this end, we employ again a variant of the quark-meson model with two quark flavors and $N_{\mathrm{c}} = 3$ colors but we now take fluctuation effects into account to analyze the effect of a finite cubic periodic box on the dynamics of this model. To be specific, the classical action underlying our studies may be viewed as an extension of the action (20) and reads

$$ S = \int d^4x \left\{ \bar{q}\left(\partial\!\!\!/ + \tfrac{1}{2}\bar{h}(\sigma + \mathrm{i}\vec{\tau}\cdot\vec{\pi}\gamma_5)\right)q + \tfrac{1}{2}(\partial_\mu\phi)^2 + U(\phi^2) - \bar{c}\sigma \right\}. \tag{57} $$

Compared to our previous studies, we allow for a term linear in the $\sigma$-field. The latter breaks explicitly the chiral symmetry. The associated parameter $\bar{c}$ is related to the quark mass through a combination of the couplings of this model, see below. The inclusion of an explicit quark mass is now essential as we aim at a study of the effect of a finite cubic periodic box on the dynamics of the model [97, 98].

In the following we shall compute the effective action in the local potential approximation where a possible space dependence of the expectation value of the scalar fields is not taken into account and the wave-function renormalizations of the fields are considered to be constant. Moreover, as also done in the studies presented in the previous subsections, we shall assume that the Yukawa coupling $\bar{h}$ does not depend on the RG scale $k$, i.e. $\bar{h}_k \equiv \bar{h}_{\Lambda'} = \bar{h}$, with $\Lambda'$ being

the scale at which we fix the couplings of the model by means of an ansatz for the effective action $\Gamma_{\Lambda'}$. Still, we include effects beyond the mean-field approximation even within such a setting, see, e.g., Refs. [73,80] for a detailed discussion of the relation of the local potential approximation and the mean-field approximation. In any case, we only use this setting here to demonstrate how RG consistency can be ensured in approximations which are more involved than the mean-field approximation. Of course, the line of arguments detailed in Sec. 2 is very general anyhow and therefore does not depend on the underlying approximations by any means.

A differential equation for the scale-dependent effective action $\Gamma_k$ can be derived with the aid of the flow equation (7). Expanding the fields about a homogeneous background $\bar{\varphi}$ and using the 3d Litim regulator for both the quark and meson fields [99,100], we obtain [101–103]:

$$k\partial_k U_k(\bar{\varphi}^2) = k\partial_k\left(\lim_{T\to 0}\frac{T}{V}\Gamma_k[\bar{\varphi}]\right) \tag{58}$$

with $V = L^3$ and

$$k\partial_k U_k(\bar{\varphi}^2) = \frac{k^4}{2}\left(-\frac{24}{\sqrt{k^2 + \frac{1}{4}\bar{h}^2\bar{\varphi}^2}} + \frac{3}{\sqrt{k^2 + 2U_k'}} + \frac{1}{\sqrt{k^2 + 2U_k' + 4\bar{\varphi}^2U_k''}}\right)\mathcal{B}(kL). \tag{59}$$

Here, $\bar{\varphi}$ denotes the homogeneous background of the scalar fields and the primes denote derivatives with respect to $\bar{\varphi}^2$. Note that the parameter $\bar{c}$ measuring the explicit breaking of the chiral symmetry does not depend on the RG scale $k$. Thus, we have

$$U_k(\bar{\varphi}^2) - \bar{c}\bar{\sigma} = \lim_{T\to 0}\frac{T}{V}\Gamma_k[\bar{\varphi}] \tag{60}$$

in our present approximation where $\bar{\sigma}$ denotes the zeroth component of the field vector $\bar{\varphi}$.

The first term on the right-hand side of the flow equation (59) is associated with the quark degrees of freedom. The second and the third term represent contributions from the mesonic modes. By dropping the latter two contributions to the RG flow of the effective action, we simply recover the mean-field effective action as already discussed above for the 3d Litim regulator in the infinite-volume limit. The explicit dependence of the RG flow on the finite periodic cubic volume $V = L^3$ is encoded in the momentum-modes counting function $\mathcal{B}$:

$$\mathcal{B}(kL) = \frac{1}{(kL)^3}\sum_{\vec{n}\in\mathbb{Z}^3}\theta\left((kL)^2 - (2\pi\vec{n})^2\right), \tag{61}$$

where $\vec{n}$ labels a three-dimensional vector of integers. In the limit $L\to\infty$, we have $\mathcal{B}(kL)\to 1/(6\pi^2)$. Thus, the flow equation (59) agrees identically with the known flow equation for the scale-dependent effective action in the local potential approximation in the infinite-volume limit [71,104], as it should be. We note that, for finite $L$, the right-hand side of the flow equation (59) is discontinuous for the Litim regulator. However, this does not cause any conceptional problem. In fact, the resulting effective action is still continuous as a function of $k$. For more detailed discussions of the properties of RG flows of finite systems with this regulator, we refer the reader to Refs. [101–103,105].

In order to compute the scale-dependent effective action $\Gamma_k$ in our present illustrational study, we shall now parametrize $U_k$ as follows:

$$U_k(\bar{\varphi}^2) = \frac{1}{2}\bar{m}_k^2\left(\bar{\varphi}^2 - \bar{\varphi}_{0,k}^2\right) + \frac{1}{4}\bar{\lambda}_k\left(\bar{\varphi}^2 - \bar{\varphi}_{0,k}^2\right)^2. \tag{62}$$

The condition

$$\frac{\partial}{\partial\bar{\sigma}}\left(\lim_{T\to 0}\frac{T}{V}\Gamma_k[\bar{\varphi}]\right)\Bigg|_{\bar{\varphi}=\bar{\varphi}_{0,k}} \overset{!}{=} 0 \tag{63}$$

then ensures that the effective action is always expanded about the actual physical ground state by relating the couplings $\bar{m}_k^2$ and $\bar{\varphi}_{0,k}$ [106]:

$$\bar{m}_k^2\bar{\varphi}_{0,k} = \bar{c}\,. \tag{64}$$

Thus, the RG flows of the two scale-dependent couplings $\bar{\varphi}_{0,k}$ and $\bar{\lambda}_k$ parametrize the RG flow of the effective action in our present approximation. The flow equations for $\bar{\varphi}_{0,k}$ and $\bar{\lambda}_k$ can be obtained by expanding the flow equation (59) about $\bar{\varphi}_{0,k}$ and then projecting it onto the logarithmic scale-derivative of the ansatz (62). The resulting flow equations can then be solved by specifying the values of the two couplings at some scale $k = \Lambda'$. For example, the parameters $\bar{\varphi}_{0,\Lambda'}$ and $\bar{\lambda}_{\Lambda'}$ may be chosen such that the physical values of a given set of low-energy observables are recovered in the long-range limit $k \to 0$.

Let us now discuss how RG consistency can be ensured in the present setting. Following our general discussion in Sec. 2, this requires a suitable adaption of the effective action $\Gamma_\Lambda$ at the scale $k = \Lambda > \Lambda'$. In the infinite-volume limit, the effective action $\Gamma_\Lambda$ at the scale $k = \Lambda$ is obtained from the given effective action $\Gamma_{\Lambda'}$ at the scale $\Lambda'$ by solving the flow equation (59) from $k = \Lambda'$ up to the scale $k = \Lambda > \Lambda'$. This ensures RG consistency, see also Eq. (14) and the related discussion. In the present case, to be specific, this corresponds to solving the set of coupled flow equations for $\bar{\varphi}_{0,k}$ and $\bar{\lambda}_k$ from $k = \Lambda'$ up to the scale $k = \Lambda > \Lambda'$ with given values for the two couplings at the scale $k = \Lambda'$. The effective action $\Gamma = \Gamma_{k\to 0}$ then does not depend on our choice for $\Lambda$, $\Lambda\partial_\Lambda\Gamma \to 0$, implying that the values for the low-energy observables, such as the constituent quark mass $m_q$, the pion decay constant $f_\pi$, and the pion mass $m_\pi$ do not depend on our actual choice for $\Lambda$.

Of course, we also would like to ensure that the RG-consistency condition is satisfied in our study of finite-volume effects. To discuss this issue further, we shall first assume that the parameters of our model (i.e. the values of $\bar{\varphi}_{0,k}$ and $\bar{\lambda}_k$ at the scale $k = \Lambda'$ as well as $\bar{c}$) have been fixed in the infinite-volume limit as discussed above. RG consistency for $1/L > 0$ can now be ensured by fixing the two scale-dependent couplings $\bar{\varphi}_{0,k}$ and $\bar{\lambda}_k$ at a scale $\Lambda > \Lambda'$ in such a way that RG consistency is still ensured in the infinite-volume limit. If $\Lambda$ has been chosen sufficiently large, i.e. $1/(\Lambda L)$ is sufficiently small, then the effective action $\Gamma$ and therefore also the physical observables become independent of $\Lambda$, i.e. $\Lambda\partial_\Lambda\Gamma \to 0$, even for $1/L > 0$. Note that the value for $\Lambda$ effectively ensuring RG consistency for a given box size depends on the regularization scheme.

With an RG-consistent effective action at hand, we may now compute physical observables, such as the pressure $P$ of the system,

$$P = -\frac{\partial}{\partial V}\left(\lim_{T\to 0} T\,\Gamma[\bar{\varphi}_0]\right), \tag{65}$$

as we have done it in the previous subsections.[5] However, since we only aim at an illustration of how RG consistency is ensured in a study beyond the mean-field approximation, we shall only discuss the volume dependence of the simplest physical observable that can be extracted from the effective action in our present setting, namely the position of its minimum in the long-range limit, i.e. $f_\pi \equiv |\bar{\varphi}_0| = \lim_{k\to 0}\bar{\varphi}_{0,k}$. To this end, we need to fix the initial conditions

---

[5]Note that the derivative of the effective action with respect to the volume is trivial in case of infinite-volume studies, see, e.g., Eq. (45). In the presence of a finite volume, the computation is more involved as the volume dependence of the couplings and $\bar{\varphi}_0$ has to be taken into account as well.

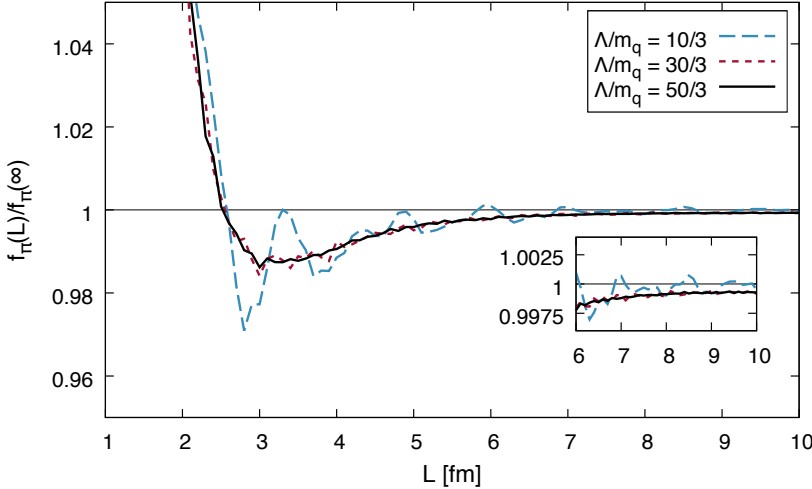

Figure 5: Normalized pion decay constant $f_\pi(L)/f_\pi(\infty)$ as a function of the box size $L$ computed with the 3$d$ Litim cutoff for $\Lambda/m_q \approx 10/3, 30/3, 50/3$, where $m_q \equiv m_q(\infty) \approx 0.300$ GeV and $f_\pi \equiv f_\pi(\infty) \approx 0.092$ GeV. For increasing $\Lambda L$, the pion decay constant $f_\pi(L)/f_\pi(\infty)$ is continuously "smoothened". In any case, deviations of the results for $\Lambda/m_q = 10/3$ from those for $\Lambda/m_q = 50/3$ are found to be on the 1% level at most. Note that the effective actions associated with the different values of $\Lambda$ agree identically in the infinite-volume limit by construction, i.e. the RG-consistency condition (1) is exactly satisfied in this limit.

for the couplings $\bar{\varphi}_{0,k}$ and $\bar{\lambda}_k$ at some scale $\Lambda'$ (as well as the parameter $\bar{c}$) which corresponds to fixing the effective action at this scale. Here, we choose the parameters such that the physical values of a given set of low-energy observables are recovered from our ansatz for the effective action $\Gamma$ in the infinite-volume limit. To be specific, the parameters are determined such that we have $m_q = \frac{1}{2}\bar{h}\bar{\varphi}_0 \approx 0.300$ GeV, $f_\pi = 2m_q/\bar{h} \approx 0.092$ GeV, and $m_\pi \approx 0.138$ GeV, in accordance with chiral perturbation theory [107]. These values are obtained by choosing $\bar{\varphi}_{0,\Lambda'}/m_q \approx 1.08 \cdot 10^{-2}$, $\bar{\lambda}_{\Lambda'} \approx 50.8$, $\bar{h} \approx 6.52$, $\bar{c}/m_q^3 \approx 6.48 \cdot 10^{-2}$ at the scale $\Lambda'/m_q \approx 10/3$.

In Fig. 5, we show our results for the pion decay constant as a function of $L$ as obtained from calculations with $\Lambda/m_q \equiv \Lambda'/m_q \approx 10/3$ as well as for $\Lambda/m_q \approx 30/3$ and $\Lambda/m_q \approx 50/3$. For $\Lambda = \Lambda'$, the pion decay constant seems to exhibit a pathological behavior when the volume is decreased. For increasing box size, however, the dependence of $f_\pi$ on $L$ is continuously "smoothened" and $f_\pi$ eventually approaches its value in the infinite-volume limit. This overall behavior of the pion decay constant does not come unexpected and can be traced back to the behavior of the momentum-modes counting function (61). The non-analytic form of the latter originates from the use of a non-analytic regulator function in our study. With such regulators, the presence of a momentum cutoff becomes very evident. Indeed, we observe that the seemingly pathological behavior is continuously "smoothened" when $\Lambda$ is increased in an RG-consistent manner. The latter corresponds to increasing the dimensionless quantity $\Lambda L$ and therefore also explains why this behavior of $f_\pi$ observed for, e.g., $\Lambda/m_q \approx 10/3$ and small box sizes goes away when the box size is increased. In particular, we observe that the results for the pion decay constant as a function of $L$ converge when $\Lambda$ is increased. In any case, we find that the deviations of the results for $\Lambda/m_q \approx 10/3$ from those for $\Lambda/m_q \approx 50/3$ are on the 1% level at most for the range of box sizes shown in Fig. 5. For a discussion of the general behavior of the pion decay constant as a function of the box size, we refer the reader to Ref. [108] where also qualitative comparisons to lattice QCD calculations [109,110] of related quantities can be found.

Note that in practice it may be advantageous – although not necessarily required – to employ analytic regulators which lead to an exponential suppression of cutoff effects when $\Lambda L$ is increased rather than a polynomial suppression as in the case of non-analytic regulators. Here, the strength of the exponential suppression is related to the first pole or cut in the complex plane introduced by the regulator. It is not necessarily the typical thermal damping in the presence of a mass. The latter decay is indeed only seen for $3d$ regulators, see Ref. [105] for a detailed discussion. In this context, we emphasize again that the meaning of the actual value of the cutoff $\Lambda$ is ambiguous without referring to the employed regularization scheme, see our discussion in Sec. 2. Indeed, the values of the momentum cutoff $\Lambda$ associated with analytic regulators used in the past to study fluctuation effects in finite volumes [97, 98, 105, 108, 111, 112] effectively correspond to larger values of $\Lambda$ in case of typically employed non-analytic regulators, such as the sharp or Litim regulator.

We would like to close our discussion with a word of caution regarding the construction of UV completions of low-energy models. In contradistinction to the mean-field studies in Subsecs. 3.1-3.3, the UV cutoff scale $\Lambda$ cannot be pushed to arbitrarily large values in the present case where we have taken into account fluctuation effects. This can be traced back to the fact that the mesonic fluctuations induce a Landau-pole instability at large momentum scales, irrespective of the employed regulator. For the simple quartic approximation of the effective action considered here, this instability occurs at comparatively large scales $\Lambda > 5\,\text{GeV}$ (for the $3d$ Litim regulator). However, the position of this instability is shifted to smaller momentum scales when corrections of higher order are included. It may then not be possible anymore to construct an RG-consistent effective action for the range of parameters of interest. In such a situation, a suitable workaround to preserve RG consistency at least approximately may be obtained by simply dropping the mesonic fluctuations in the construction of the UV completion, i.e. by only employing the corresponding mean-field UV completion simply constructed from the fully field-dependent quark loop as detailed in Subsecs. 3.1-3.3. In practice, this may already suffice to reduce cutoff artefacts to a large extent in studies beyond the mean-field approximation.

## 4 Conclusions

In the present work we have discussed the concept of RG consistency in the context of model studies, with an emphasis on studies in the presence of external parameters. In general, RG consistency requires that the effective action $\Gamma$ of a given theory does not depend on the cutoff scale $\Lambda$, i.e. $\Lambda\partial_\Lambda\Gamma = 0$, also in the presence of external parameters. After a detailed general discussion of RG consistency in Sec. 2, we have given an illustrative discussion of RG consistency in mean-field studies of a quark-meson model in the vacuum limit, a diquark model at finite density, and a quark-meson-diquark model at finite temperature and density. We note that in the latter two cases, we had to take into account the renormalization of the diquark chemical potential to ensure RG consistency. Moreover, we discussed RG consistency in studies of finite-volume effects by considering the quark-meson model beyond the mean-field approximation.

For regularization schemes and values of the cutoff scale $\Lambda$ as widely employed in mean-field studies of QCD models, our illustrational studies already suggest that "cutoff contaminations" of physical observables can be significant. Indeed, for the zero-temperature pressure of our quark-meson-diquark model, we found corrections of up to 30% in the considered range for the quark chemical potential. However, it is not only the computation of the pressure that suffers from "cutoff contaminations". For example, the critical temperature of our quark-meson-diquark model at $\mu = 0$ is lowered by about 10% when we take into account cutoff

corrections enforced by the RG-consistency condition (1). In general, such corrections do not necessarily only lead to a decrease of the critical temperature. In fact, in the regime governed by diquark condensation, cutoff corrections rather tend to increase it. To be specific, the critical temperature is increased by about 30% at $\mu/m_q = 4/3$ (with $m_q \approx 0.300\,\text{GeV}$) and already by more than 100% at $\mu/m_q = 2$ compared to the results from a conventional mean-field study. Thus, the implementation of RG consistency appears to be very relevant in the high-density regime of our QCD models. For example, the associated corrections may significantly alter the presently available equations of state of dense strong-interaction matter as relevant for astrophysical applications [96]. In any case, our illustrational studies show clearly that it is of phenomenological relevance to ensure RG consistency in general model studies, even in the mean-field approximation.

## Acknowledgments

The authors as members of the fQCD collaboration [113] would like to thank the other members of this collaboration for discussions. J.B. acknowledges support by the DFG under grant BR 4005/4-1 (Heisenberg program) and by HIC for FAIR within the LOEWE program of the State of Hesse. Moreover, this work is funded by the Deutsche Forschungsgemeinschaft (DFG, German Research Foundation) – Projektnummer 279384907 – SFB 1245.

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
