# Peer review of "Renormalization group consistency and low-energy effective theories"

_SciPost Physics, doi:SciPost Phys. 6, 056 (2019)_

## Round 1 · Referee Report · Anonymous · 2019-4-6

Strengths

1-very well written, clearly structured

2-understandable for a rather wide audience of people interested in effective field theories, even though examples only QCD related

3-will probably trigger further studies of RG consistency in other models

Weaknesses

-not so many really new results, but still extremely useful compilation with some relevant new calculations

Report

In this paper the validity and consistency of effective field theories is discussed from a functional renormalization proup perspective. Renormalization group consistency is applied to a variety of effective models in the context of QCD at finite temperature and density. The examples suggest that the application of RG consistency could lead to significant changes in predictions from a wide range of other effective field theories.

The manuscript is very well written and provides an extremely useful introduction to the subject for a rather general audience, even though the applications discussed focus only on QCD related examples. The presentation is technically sound and the calculations well illustrated and informative.

The authors are leading experts in the field and the quality of the investigation is very high. Though the manuscript has somewhat the character of an introductory review, the example calculations provide some very interesting new results that will certainly trigger further studies in this field of research.

I recommend publication of this very nice paper without changes.

Requested changes

- none

---

## Round 1 · Referee Report · Anonymous · 2019-4-10

Strengths

Important subject

Clear and correct results

Well written

Weaknesses

It takes perhaps a little long before the reader comes to the interesting figures that make the aim of the authors very clear. Perhaps an earlier similar figure for the quark meson model, perhaps even before the formal discussion, could be helpful.

Report

This is a very interesting and valid paper on the question of cutoff dependence in low -energy effective field theories. The issue is mainly tackled within functional renormalisation, but has validity quite more generally.
Expectation values of observables depend on some microscopic couplings and on the cutoff for the momentum integrations within the effective theory. Ideally, the cutoff just denotes the transition scale from some more fundamental theory to the effective theory, and observables should not depend on it. In practice, this dependence is often rather strong, such that the cutoff appears as a parameter of the effective theory. The present paper proposes renormalisation group consisteny as powerful method to reduce the cutoff dependence. They show what can be achieved with well chosen practical examples.
I recommend publication

Requested changes

none

---

## Editorial Decision

published